# Heart Rate Variability Measurements Across the Menstrual Cycle and Oral Contraceptive Phases in Two Olympian Female Swimmers: A Case Report

**DOI:** 10.3390/sports13060185

**Published:** 2025-06-12

**Authors:** Marine Dupuit, Kilian Barlier, Benjamin Tranchard, Jean-François Toussaint, Juliana Antero, Robin Pla

**Affiliations:** 1UMR 7329—IRMES, INSEP, Institut de Recherche Médicale et d’Épidémiologie du Sport, Université Paris Cité, F-75012 Paris, France; kilian.barlier@grand-insep.fr (K.B.); jean-francois.toussaint@aphp.fr (J.-F.T.); juliana.antero@insep.fr (J.A.); robin.pla@ffnatation.fr (R.P.); 2Fédération Française de Natation, 92110 Clichy, France; benjamin.tranchard@ffnatation.fr; 3Centre d’Investigation en Médecine du Sport, CIMS Hôtel-Dieu, Assistance Publique—Hôpitaux de Paris, 75004 Paris, France

**Keywords:** heart rate variability, menstrual cycle, oral contraception, elite female athlete, swimming

## Abstract

The heart rate variability (HRV), influenced by female sex hormone fluctuations, is an indicator of athletes’ adaptation. This case study explores HRV responses over 18 months across a natural menstrual cycle (MC) and during oral contraceptive (OC) use in two Olympic female swimmers. HRV measurements—including mean heart rate (HR); root mean square of successive differences (RMSSD); and frequency-domain indices—were collected at rest in supine (SU) and standing (ST) positions across two competitive seasons. Nocturnal HR and RMSSD were assessed using the Ōura^®^ ring. MC and OC phases were identified through specific tracking, and training load was controlled. In both athletes, resting HR was lower during bleeding phases, increasing from menstruation to the luteal phase (MC) and from withdrawal to active pill phases (OC). In the ST position, RMSSD was higher but decreased throughout the phases. Nocturnal measurements confirmed these trends. Overall, findings suggest a phase-related parasympathetic overactivity shift. This study provides novel insights into HRV responses across hormonal cycles in elite female athletes, which present unique characteristics. Such monitoring tools may support a data-informed approach to guide and periodize training more effectively.

## 1. Introduction

Numerous physiologists and researchers have investigated the assessment of heart rate variability (HRV) in order to evaluate the activity of the autonomic nervous system in elite athletes [1,2], including swimming [3,4,5,6]. HRV is one of the most informative and easiest tools to monitor athletes’ adaptations. Recently, some studies stated that HRV-guided training may be more effective than predefined training to improve cardiovascular adaptations and performance [7,8,9]. The HRV helps to understand the activity of ortho- and parasympathetic systems, the two branches of the autonomous nervous system (ANS). Many authors described the possibility to identify elite athletes’ individual responses to training load and physiological stress during specific periods [4,10,11,12].

Regarding individual characteristics impacting HRV measures (e.g., age, physical activity, body mass index, etc. [13]), one of these could be the natural menstrual cycle (MC) with its ovarian hormone fluctuations, especially estradiol (E2) and progesterone (P4) [14,15]. While inter- and intra-individual MC variations can be observed, a typical MC ranges from 21 to 35 days and is usually divided into 2 phases: Follicular (FP), from menses to ovulation day, and luteal (LP), past ovulation [16]. These phases are characterized by specific hormonal fluctuations, which also provide a six-subphase classification [17]. E2 and P4 affect a wide array of psychological and physiological functions. Some of these functions are also linked to the ANS regulation; the MC may induce variations in HRV response across the cycle [14]. In eumenorrheic women, recent meta-analyses and systematic reviews [14,18] have suggested a parasympathetic predominance during the FP, which declines across the MC, and, in contrast, a sympathetic predominance in the progesterone-dominated phase (i.e., LP). A high percentage of female athletes use hormonal contraception, especially combined monophasic oral contraceptive (OC) pills [19]. Combined OC downregulates endogenous E2 and P4 production by providing an active pill with consistent concentrations of synthetic hormones, ethinyl estradiol and progestin, for 21 days. Therefore, HRV responses in OC users could differ from those with natural MC but also between the active pill-taking phase (APP) and placebo or withdrawal phase (WP) [20,21,22].

Among studies investigating HRV in females, publications centered on elite female athletes are scarce, and few of them considered menstrual status. Moreover, rare conclusions must be interpreted with caution because of some inconsistencies in study design and methods on both HRV and MC assessments. For instance, HRV test conditions such as positions (e.g., supine, sit, stand-up), recording time, time of day (diurnal or nocturnal), devices used (e.g., heart rate monitor, ring, whistle, watch, phone, etc.), and selected domain analysis (e.g., time, frequency, fractal) [23] may contribute to the lack of clarity on HRV across the natural MC and OC phases. HRV trends are also generally obtained from data collected only on one or two cycles. Moreover, one- to three-step methods are used to determine MC phases [16], and the MC phases division varies from the simplest (two phases) to the highly detailed (seven phases) [14]. In addition, while a 10 min orthostatic test is essential for precise athlete monitoring [24], it might be a high participant burden leading to poor compliance [25]. In this context, wearable devices, such as the Ōura^®^ ring, could be an interesting alternative to avoid this issue.

To our knowledge, there is no study reporting HRV measurements over a long period of monitoring in elite female athletes considering their hormonal cycles. The assessment of HRV across the menstrual cycle may provide valuable insights into ANS activity during the different phases of the cycle. Such measurements could help elite athletes and their staff manage training loads and recovery strategies more effectively. Thus, the present case study aimed to explore the resting and nocturnal HRV, respectively assessed via orthostatic test and Ōura^®^ ring, across natural MC and OC phases in two Olympian female swimmers. We hypothesized that a rebound in parasympathetic activity might be observed during the menstrual phase and during the withdrawal phase of combined OC use.

## 2. Materials and Methods

### 2.1. Subjects

This study included two female Olympian swimmers who reached the Top 10 during the 2024 Olympic Games and were both European champions throughout their careers (Tier 5, world-class athletes [26]). Athlete 1 is 30 years old and naturally menstruates (176 cm, 64 kg). Athlete 2 is 27 years old and uses a monophasic oral contraceptive pill (182 cm, 77 kg).

HRV monitoring (detailed below) is routinely included in athlete follow-up by the French Swimming Federation. Menstrual status monitoring (detailed below) was part of the Empow’Her project as previously described [27,28,29,30,31,32], which was approved by the Institutional Ethics Committee (IRB00012476-2022-03-11-206). The data collection process adhered to the code of ethics outlined by the World Medical Association (Declaration of Helsinki) and received a certificate of compliance from the Commission Nationale Informatique et Libertés (CNIL-2221532 v0). The two athletes gave their written informed consent for data recording and utilization in the context of training and research.

### 2.2. Study Design

On-field longitudinal case study based on daily monitoring.

Figure 1 represents the study design and the timeline of data collection for each athlete. For Athlete 1, the menstrual status, training load, and HRV data recorded via orthostatic test were monitored from the end of February 2023 to the end of the Olympic Games in August 2024. Her HRV night measurements have been recorded since November 2023, when she received the Ōura^®^ ring (Ōura Health, Oulu, Finland, 3rd generation). For Athlete 2, two 28-day oral contraceptive (OC) cycles were also retrospectively identified to align with the period during which she was wearing the Ōura^®^ device, and all HRV data (orthostatic test and night measurements) were recorded.

### 2.3. HRV Monitoring

HRV recordings were collected using two measurement methods: (1) Orthostatic test just after awakening in supine (SU) and standing (ST) positions and (2) nocturnal, during sleep.

#### 2.3.1. Orthostatic Test (HRV After Awakening)

The HRV protocol test was performed as described by Schmitt et al. (2013) [33]. Each test was conducted in the morning just after awakening. Swimmers have performed at least two HRV tests weekly for the last 4 years. The same routine was used for every test recording. The test lasted 5 min in the SU position, then 5 min in the ST position. RR intervals were recorded with an HR belt (Polar H10, Polar Electro Oy, Kempele, Finland) and transmitted by Bluetooth using a smartphone application (Elite HRV, Gloucester, MA, USA). RR interval recordings were analyzed with the last 240 s of each position [25]. Breathing frequency was not guided. All RR recordings were visually inspected for stationarity and corrected for artifacts and ectopic beats. The interpolated RR interval series were used to compute HRV spectra by employing a Fast-Fourier Transformation with Welch’s periodogram method. This definition ensures that the HRV spectral parameters were estimated from a single window containing the whole 240 s period of recording. Three time-domain indices were used for the HRV analysis: HR, root mean square of successive difference (RMSSD), and standard deviation of NN intervals (SDNN). Two HRV primary variables were calculated from the frequential domain with Fast Fourier Transformation: Low frequency (LF, 0.04–0.15 Hz) and high frequency (HF, 0.15–0.4 Hz). Then, from these metrics, the sum of LF+HF, HF/HR ratio in SU position, and LF/HF in ST position were used for further analysis.

#### 2.3.2. Night Measurements

To assess HRV measurements during the night, the athletes wore a biometric ring (Ōura Health, Oulu, Finland, 3rd generation). Athlete 1 received the device in November 2023, and Athlete 2 in December 2022. The athletes were asked to wear the ring on the index finger of their dominant hand every night and as much as possible during the day, except for training sessions. They were also asked to open the application every morning in order to synchronize the data. The sensor detects HR and HRV (i.e., RMSSD) from the finger optical pulse waveform (i.e., infrared photoplethysmography), showing a high agreement compared to gold-standard electrocardiography for night measurements [34].

### 2.4. Menstrual Status Monitoring and Phase Determination

As previously described [27,28], it relied on a menstrual diary. The two athletes had to complete daily an online application indicating the start and the end of her menstruation (Athlete 1) or her placebo pill (Athlete 2). Then, phases were defined depending on their menstrual status.

For Athlete 1 with a natural MC, phases were determined using “two-step” methods [35] combined with salivary hormone dosages. She completed a menstrual diary, allowing her to determine menstruation phase and cycle length (step one). From the third to sixth cycles, she used urinary ovulation strip tests (Premom, Tianjin, China) each morning starting from day ten until a positive result (i.e., luteinizing hormone surge) to detect ovulation day (step two). Ovulation generally occurs 12 to 24 h after the luteinizing hormone surge detected by the test [36]. The follicular phase (FP) was defined as the time from the first day of bleeding to ovulation, and the luteal phase (LP) from the day after ovulation up to the day preceding the next bleeding. FP and LP phases were divided into 3 sub-phases according to hormonal fluctuations: Menstruation (i.e., low levels of E2 and P4 at the beginning of the follicular phase), mid-FP and late-FP (i.e., an E2 rise and peak during the FP before ovulation), early-LP and mid-LP (i.e., a P4 rise and peak after ovulation concomitantly with a second lesser E2 peak), and late-LP (i.e., an E2 and P4 falling) [17,28,37]. In addition, saliva samples, a valid method to identify normal from abnormal hormonal cycles [38,39], were performed on the third cycle as described by Lafitte et al. (2024) to measure 17β-estradiol (E2), progesterone (P4), and free testosterone concentrations (pg.mL^−1^) [31]. The distal body temperature from the Ōura^®^ ring was also used to detect ovulation day [40].

For Athlete 2 using OC pills (Minidrill^®^), her cycles were divided into two phases according to the pills’ usage: The withdrawal phase (WP) of 7 days (i.e., placebo, pause) and the active pill-taking phase (APP) of 21 days 28.

### 2.5. Training Load

Training load was quantified over the monitoring period for each athlete according to their coach’s methodology.

For Athlete 1, training volume was collected as total distance swum in meters per session. The swimming distance was also divided into 4 intensity zones and is also recorded and converted into percent of total distance: (1) Under her first ventilatory threshold (<75% maximal HR), (2) between the first and the second ventilatory threshold (between 75 and 88% maximal HR), (3) above the second ventilatory threshold (>88% maximal HR), or (4) corresponding to anaerobic alactic efforts (e.g., short sprint). Training load was expressed in arbitrary units (au), using the first Banister’s TRIMP calculation (1975) [41]. Distances in each of the 4 intensity zones were multiplied by a corresponding weighted intensity coefficient (from 1 to 4), which placed greater weighting on higher intensities.

For Athlete 2, training volume in kilometers and intensity were collected. Intensity was measured with the rating of perceived exertion (RPE) method using the Borg scale (modified version). Training load is represented in arbitrary units as the RPE session (sRPE) [42]: sRPE = training volume (km) × RPE.

### 2.6. Statistical Analysis

The normality of the distributions was tested with a Shapiro-Wilk test at the 0.05 threshold. As the majority of variables were not normally distributed, and for the sake of consistency, non-parametric tests with three significance levels (0.05, 0.001, 0.0001) were used on all variables to identify median differences in HRV variables depending on the phase of the cycle. A different methodology was used for each athlete due to the differences in menstrual status between them (natural MC for Athlete 1 and hormonal OC for Athlete 2). For Athlete 1, the Kruskal–Wallis test was used on each variable to point out median differences depending on menstrual sub phases (menstruation, mid-FP, late-FP, early-LP, mid-LP, and late-LP). A post-hoc Wilcoxon test was used to compare the phases in pairs and identify which ones showed significant differences in terms of median. For Athlete 2, the Wilcoxon test was used on each variable to identify median differences on HRV variables depending on the phases (WP vs. APP). The mean training load in each phase was compared between the six subphases with the Kruskal–Wallis test for Athlete 1 and between the two OC phases with the Wilcoxon test for Athlete 2.

## 3. Results

### 3.1. Athlete 1

#### 3.1.1. Menstrual Cycle

A total of 18 regular MCs varying between 28 and 34 days (29.8 ± 1.9 days) were monitored across the following period. Ovulation tests were performed in 4 non-consecutive cycles. For these cycles, phases were determined from the confirmed ovulation day. Saliva samples, collected during the third MC (Appendix A), showed expected E2 and P4 fluctuation and confirmed ovulatory cycle [38,39]. Based on these tests and the variation of distal body temperature from the Ōura^®^ ring, we assumed all cycles were ovulatory [40]. For other cycles (i.e., with no ovulation test and salivary hormonal sample), phase determination was based on detected ovulation day via the Ōura^®^ ring or on the estimated ovulation day as previously described [28].

#### 3.1.2. Training Load

The mean training load in arbitrary units was not significantly different between phases (menstruation: 17,115 ± 7430; mid-FP: 16,734 ± 8105; late-FP: 17,500 ± 7388; early-LP: 17,094 ± 6883; mid-LP: 17,131 ± 7326; and late-LP: 17,985 ± 7624, *p* = 0.91). Her weekly training load as a percentage of the maximum week is presented in Appendix A.

#### 3.1.3. Orthostatic Awaking Test

A total of n = 207 tests were recorded, with n = 35 in the menstruation phase, n = 49 in mid-FP, n = 30 in late-FP, n = 26 in early-LP, n = 35 in mid-LP, and n = 32 in late-LP.

HR mean significantly increases across the MC from the menstruation to LP (*p* < 0.0001) in both SU and ST positions.

In the SU position, HR mean is significantly lower during menstruation compared to early-, mid-, and late-LP (*p* = 0.001, *p* = 0.003, and *p* = 0.001, respectively, Figure 2A) and during mid-FP compared to early- and late-LP (*p* = 0.048 and *p* = 0.038, respectively, Figure 2A).

The same variations are observed in ST position with a significantly lower HR mean during menstruation versus early-, mid-, and late-LP plus late-FP (*p* = 0.009, *p* = 0.01, *p* < 0.0001, and *p* = 0.034, respectively; Figure 2B). HR mean is also significantly lower in mid-FP versus late-LP (*p* = 0.023, Figure 2B).

In the SU position, no difference is found on RMSSD (Figure 2C), SDNN, and LF, whereas HF tends to differ with higher values in FP compared to LP (*p* = 0.09, Table 1). Concerning ratio, the same trend is observed on HF/HR ratio (*p* = 0.08, Figure 3A).

In the ST position, RMSSD decreases across the MC (*p* < 0.001, Figure 2D). It is significantly lower in late-LP than menstruation, mid- and late-FP (*p* = 0.005, *p* = 0.045, *p* = 0.025, respectively, Figure 2D). The same variations are observed on HF (*p* = 0.010, *p* = 0.023, *p* = 0.034, respectively, Table 1). No difference is found on SDNN and LF (*p* > 0.05, Table 1). The ratio LF/HF significantly varies across the MC (*p* = 0.040), with lower values in menstruation compared to late-LP (*p* = 0.015, Figure 3B).

#### 3.1.4. Night Measurements

Across the 39 weeks of Ōura^®^ ring data collection, n = 250 night observations were recorded, corresponding to 89.0% adherence. Their occurrences in each phase of the nine covered MCs varied between 25 and 50: menstruation n = 43, mid-FP n = 59, late-FP n = 31, early-LP n = 25, mid-LP n = 42, and late-LP n = 50.

HR mean significantly increases from menstruation (*p* < 0.0001). HR mean is significantly lower during menstruation compared to early-, mid-, and late-LP and tends to be lower than late-FP (*p* < 0.001 for each phase and *p* = 0.062, Figure 2E). HR mean is also lower during mid-FP compared to late-FP and early-, mid-, and late-LP (*p* = 0.037 and *p* < 0.0001 for each LP, Figure 2E). HR mean is higher in late-LP vs. late FP (*p* < 0.0001, Figure 2E). A significant difference is observed in RMSSD across the MC (*p* < 0.0001). It is significantly higher in mid- and late-FP compared to mid- and late-LP (*p* = 0.008 and *p* = 0.005 for both, Figure 2F).

### 3.2. Athlete 2

#### 3.2.1. Oral Contraceptive Use

Across the following period, Athlete 2 has taken a total of 19 pills/tablets (Minidrill^®^). Due to forgotten pills or voluntary adjustments, OC cycles varied between 24 and 30 days (27.8 ± 1.5 days).

#### 3.2.2. Training Load

The mean training load in arbitrary units was not significantly different between the WP (14.6 ± 14.2) and the APP (9.8 ± 12.8, *p* = 0.13). Her weekly training load as a percentage of the maximum week is presented in Appendix A.

#### 3.2.3. Orthostatic Awaking Test

A total of n = 96 tests were recorded, with n = 25 tests in WP and n = 71 tests in APP.

The numbers of HRV data collected via orthostatic tests available in each phase are represented in Table 1. The HR mean is significantly lower during the WP in both SU and ST positions (*p* = 0.011 and *p* = 0.009, Figure 4A and Figure 4B, respectively). In the SU position, no difference is found on RMSSD (*p* = 0.652, Figure 4C), SDNN, absolute HF, absolute LF, and HF/HR ratio (*p* > 0.05, Table 2). In the ST position, a lower RMSSD is observed during the WP vs. APP (*p* = 0.034, Figure 4D). Absolute LF and LF/HR ratios are significantly higher during the WP (*p* = 0.040 and *p* = 0.033, Table 2). There is no difference in SDNN, absolute HF, and LF/HF ratio.

#### 3.2.4. Night Measurements

Across the 22 months of Ōura^®^ ring data collection, n = 379 night observations were recorded, corresponding to 75.3% adherence, with n = 99 occurrences in WP and n = 280 in APP. As observed via orthostatic tests, HR mean is significantly decreased in WP (*p* < 0.0001, Figure 4E), and RMSSD is significantly higher in the same phase (*p* < 0.0001, Figure 4F).

## 4. Discussion

The current case study is the first to explore the relationships between HRV parameters and MC or OC phases in two world-class female swimmers using data from orthostatic tests and night measurements across a longitudinal follow-up. Our main findings indicate a lower resting HR during the menstruation phase (natural MC) or the WP (oral contraceptive) in SU and ST positions, combined with a higher RMSSD value, particularly in the ST position. These observations seem to highlight a shift of parasympathetic overactivity during the bleeding phase in both natural MC and OC use. Furthermore, various HRV parameters appear to fluctuate across the MC.

First, we observe a lower resting HR during the menstruation phase (Athlete 1) and the WP (Athlete 2) in SU and ST positions. Specifically, resting HR significantly increases from menstruation to LP in both SU and ST positions. Conversely, RMSSD is higher in the early, mid-, and late-FP than in the LP phase (Athlete 1). In the same way, a lower resting HR in WP vs. APP and a higher RMSSD in both SU and ST positions are observed for Athlete 2. While diurnal and nocturnal HRV can differ [43], data from the Ōura ring^®^ demonstrate similar variations of resting HR and RMSSD for both athletes consistent with their respective waking tests. Altogether, these results suggest a predominance of parasympathetic activity during the bleeding phases and a decrease in vagal activity across the natural MC and between OC phases. Furthermore, the HF component, which is widely considered as a parasympathetic index [23,44], observed a similar trend to RMSSD across the natural MC. This increase in parasympathetic activity in elite athletes was already described in the literature, but it could be associated with positive adaptations to training or lead to functional overreaching [45,46].

Previous studies have examined the effects of the MC on HRV parameters in athletes, but comparisons are challenging due to variations in the phases analyzed, HRV assessment methods (i.e., domains, parameters, positions), and typically short follow-up durations of one or two cycles [14]. To your knowledge, no prior study has used orthostatic tests across a longitudinal follow-up, complicating direct comparisons with the literature. Nevertheless, most studies have reported associations between sex hormone fluctuations during the MC and changes in parasympathetic and sympathetic activities. Our findings align with Schamlenberger et al. (2019) [14], whose meta-analysis showed a significant decrease in parasympathetic activity from the FP to LP. A retrospective study analyzed nocturnal resting HR and HRV scores collected by WHOOP^®^ wearable wristbands from naturally menstruating athletes and combined hormonal contraception and progestin-only contraception users. In accordance with our results, they reported a lower resting HR and an elevated HRV in the early and mid-FP with a significant increase and decline in metrics, respectively, into the late LP in naturally menstruating athletes. Estrogens are known to centrally influence the ANS by enhancing vagal activity and suppressing sympathetic activity. In contrast, progesterone exerts an opposing effect by increasing central norepinephrine release, thereby promoting sympathetic drive [21]. Moreover, recent studies have shown that concentrations of E2 and P4 are associated with specific variations of HR and HRV metrics [14,47], supporting our findings in Athlete 1. Very few studies have investigated the HRV in OC users, particularly comparing WP and APP. Contrary to our findings, Teixeira et al. (2015) [22] found no change in resting HR and HRV components. However, results from Ahokas et al. (2023) [47], based on nocturnal HRV, are consistent with our observations. Similarly, Sims et al. (2021) [21], the only study on athletic populations, reported elevated HRV in WP with a decline at the onset of APP, though no changes in resting HR. It is worth noting that these studies differ in methodology as well as in the types and brands of contraceptive pills used. In OC users, exogenous hormone consumption downregulates endogenous hormone production to prevent ovulation [31]. During the WP, synthetic hormones are not consumed, but how the body responds to this transition remains unclear. The production and level of endogenous hormones likely depend on the half-lives of exogenous steroids. In the APP, a stable hormonal profile develops. Synthetic hormones exert their inhibitory effects on endogenous hormones, which may impact HRV, leading to a reduction in parasympathetic activity [47]. This aligns with our findings in Athlete 2. Otherwise, some studies observed an increase in the cardiovagal baroreflex sensitivity during the pre-ovulation phases [48] and during the first week of APP (i.e., low hormone phase) [49]. This might account for the elevated LF values observed in the mid- and late-FP in Athlete 1, aligning with the rise in estrogen levels during these phases. In Athlete 2, LF values, and consequently LF+HF, were significantly higher in the WP phase, when the body deprives itself of synthetic hormones and relies on endogenous estrogen production.

Increases in parasympathetic activity can reflect a variety of physiological changes. Vagal-related HRV indices, for instance, are often associated with cardiorespiratory fitness [50], underscoring positive adaptations to training [11,45]. However, elevated parasympathetic indices can also indicate overreaching, as demonstrated in studies by Bellenger et al. (2016) [51]. In the context of the MC, parasympathetic overactivity appears to result from non-training-related factors, presenting a challenge for interpretation. This complexity highlights the need for a more comprehensive approach to prescribing appropriate training for elite female athletes. To better estimate training tolerance and readiness, we strongly recommend using additional objective physiological and physical markers, such as aerobic capacity or strength tests, and subjective measures, such as RPE or fatigue questionnaires, will be appropriate to estimate the need to adapt training load and the risks of overtraining.

This case report has certain limitations inherent to its design. Various environmental factors, such as sleep, psychological factors, hydration, dietary patterns, and training load, might possibly influence HR and HRV metrics. Because of the long monitoring period, it was not feasible to control all these variables precisely. Unfortunately, it was not possible to use the same training load quantification method for both swimmers due to the different training approaches of their respective coaches. With these different quantification methods, training load did not significantly differ between their respective phases and was not adjusted according to specific phases. With our extended follow-up period, the substantial amount of collected data minimizes the influence of isolated external factors. Additionally, as elite athletes, their heightened awareness of recovery’s importance and their disciplined routines possibly mitigated these influences. A notable limitation is the MC phase determination. We relied on urinary ovulation tests (i.e., the two-step method) on only four cycles and assumed the other cycles to be ovulatory. However, this was supplemented by salivary hormonal dosages for one cycle to enhance accuracy. To our knowledge, this is the first study that monitors MC and OC cycles over such a long time in top-level elite female athletes. In addition, one of the strengths of this study is the use of two HRV assessment methods, including the orthostatic test integrated into a longitudinal follow-up.

MC or OC cycles present high interindividual variability considering personal experiences and underpinning physiology. Capturing these individual variations and using them to inform future practice is crucial, especially in elite sport. Case studies, as highlighted by Burden et al. (2024) [52], can offer valuable, precise insights into these dynamics. Our results on HR and HRV metrics across the MC and OC phases underscore the importance of collecting specific female data to tailor training load and strategy recovery. The individual physiological patterns observed in athletes may play a key role in optimizing training and recovery monitoring. The agreement between our nocturnal and awakening test in SU position observations suggests a wearable device like the Ōura^®^ ring can be a practical tool for this purpose [34,53]. However, it is important to note that such devices analyze only HR and a single HRV metric (i.e., RMSSD). This limitation may result in the omission of valuable information from the frequency domain, potentially leading to misinterpretations and suboptimal training adaptations.

## 5. Conclusions

This study provides novel insights into the relationships between HRV parameters and menstrual or OC cycle phases in elite female athletes through longitudinal monitoring. Our findings indicate a shift towards parasympathetic overactivity during bleeding phases across MC and OC use. Wearable devices like the Ōura^®^ ring can offer practical tools for monitoring HR and HRV, keeping in mind their limitation in scope, requiring complementary measures to ensure accurate training adaptations. This case study highlights the importance of considering individual physiological responses when prescribing training and emphasizes the need for personalized approaches to optimize both performance and recovery. HRV is a practical and accessible tool that can be monitored daily using recent devices. Collecting long-term HRV data may provide deeper physiological insights across various populations, including elite athletes. In particular, HRV monitoring could be especially valuable for exploring female-specific physiological characteristics.

## Figures and Tables

**Figure 1 sports-13-00185-f001:**
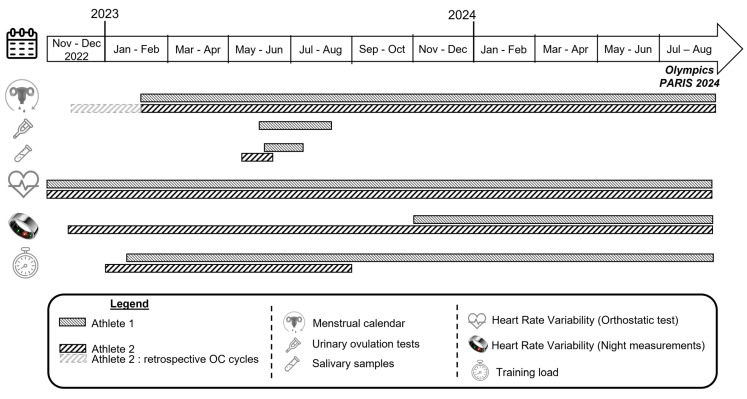
Study design and timeline of data collection.

**Figure 2 sports-13-00185-f002:**
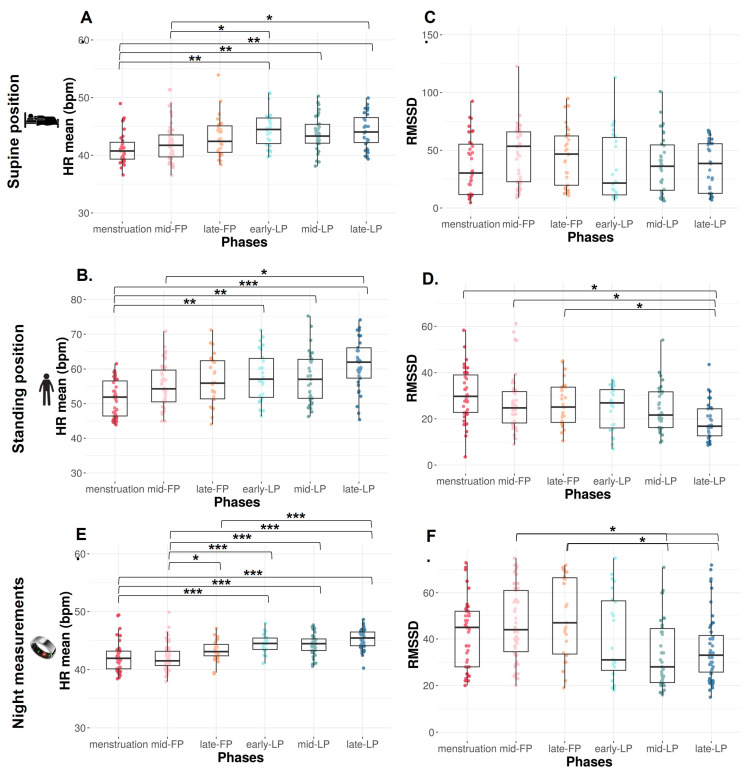
HR mean and RMSSD from orthostatic awakening test in supine position (**A**,**B**), in standing position (**C**,**D**), and from Ōura ring (**E**,**F**) for Athlete 1. Box plot represented as median and interquartile range. HR: Heart rate; RMSSD: Root of mean square of successive difference; mid-FP: Mid-follicular phase; late-FP: Late follicular phase; early-LP: Early luteal phase; mid-LP: Mid-luteal phase; late-LP: Late luteal phase. * *p* < 0.05; ** *p* < 0.001; *** *p* < 0.0001.

**Figure 3 sports-13-00185-f003:**
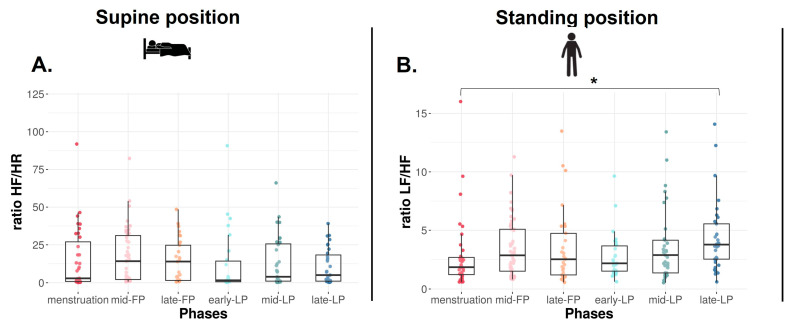
HF/HR (**A**) and LF/HF (**B**) ratios in supine and standing positions for Athlete 1. Box plot represented as median and interquartile range. HR: Heart rate; HF: High frequency; LF: Low frequency; mid-FP: Mid-follicular phase; late-FP: Late follicular phase; early-LP: Early luteal phase; mid-LP: Mid-luteal phase; late-LP: Late luteal phase. * *p* < 0.05.

**Figure 4 sports-13-00185-f004:**
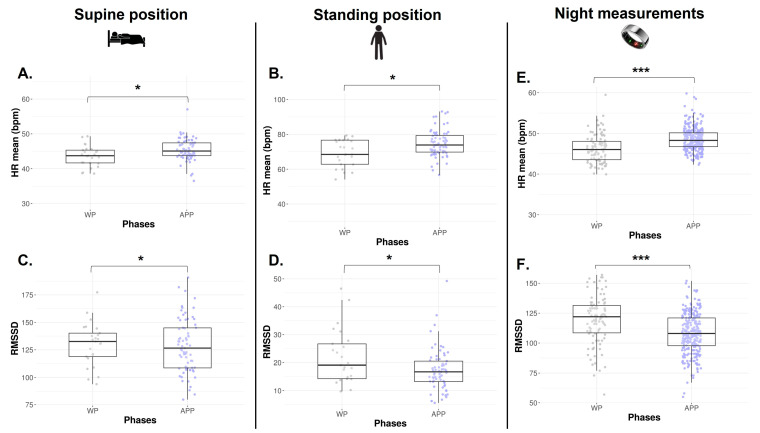
HR mean and RMSSD from orthostatic awakening test in supine position (**A**,**C**) and standing position (**B**,**D**) and from Ōura^®^ ring (**E**,**F**) for Athlete 2. Box plot represented as median and interquartile range. HR: Heart rate; RMSSD: Root of mean square of successive difference; WP: Withdrawal pill; APP: Active pill-taking phase. * *p* < 0.05; *** *p* < 0.0001.

**Table 1 sports-13-00185-t001:** Time and frequential domain parameters for Athlete 1.

Phases	Menstruation	Mid-FP	Late-FP	Early-LP	Mid-FP	Late-FP	*p*-Value
SU position							
SDNN	41 (25–64)	52 (37–71)	58 (38–67)	31 (19–75)	45 (28–62)	44 (25–76)	0.234
LF (ms^2^)	303 (42–902)	344 (99–796)	313 (87–775)	107 (51–570)	150 (48–544)	289 (88–512)	0.267
HF (ms^2^)	112 (39–1030)	576 (93–1245)	562 (71–1114)	71 (29–691)	169 (45–1030)	270 (50–906)	0.093
LF+HF (ms^2^)	446 (82–1910)	983 (195–2180)	924 (168–1676)	206 (78–1461)	474 (106–1478)	542 (142–1432)	0.157
ST position							
SDNN	55 (45–72)	52 (43–64)	59 (46–69)	60 (37–72)	49 (39–68)	48 (35–66)	0.732
LF (ms^2^)	471 (286–788)	512 (336–798)	431 (304–801)	432 (292–650)	373 (241–768)	376 (193–722)	0.391
HF (ms^2^)	278 ^a^ (128–483)	206 ^b^ (120–309)	226 ^c^ (100–396)	219 (70–328)	150 (85–324)	96 (52–193)	0.006
LF+HF (ms^2^)	844 (415–1311)	790 (492–1087)	827 (477–1079)	702 (461–921)	696 (355–919)	530 (250–96)	0.143

Values are represented as median [interquartile range]. SDNN: Standard deviation of NN intervals; VLF: Very low frequencies; LF: Low frequencies; HF: Low frequencies; SU: Supine position; ST: Standing position; mid-FP: Mid-follicular phase; late-FP: Late follicular phase; early-LP: Early luteal phase; mid-LP: Mid-luteal phase; late-LP: Late luteal phase. ^a^ menstruation vs. late-LP; ^b^ mid-FP vs. late-FP; ^c^ late-FP vs. late-LP; ^a,b,c^ *p* < 0.05.

**Table 2 sports-13-00185-t002:** Time and frequential domain parameters for Athlete 2.

Variables	Phases			Variables	Phases		
SU Position	WP	APP	*p*-Value	ST Position	WP	APP	*p*-Value
SDNN	101 (92–109)	105 (90–114)	0.413	SDNN	55 (48.02–70.18)	51 (37–61)	0.129
LF (ms^2^)	1271 (993–1652)	1471 (995–2195)	0.411	LF (ms^2^)	723 (486–1151)	579 (290–887)	0.041
HF (ms^2^)	5309 (4301–6269)	5306 (3968–6777)	0.757	HF (ms^2^)	175 (113–361)	148 (92–228)	0.127
LF+HF (ms^2^)	7012 (5488–8129)	7243 (4997–8614)	0.593	LF+HF (ms^2^)	956 (586–1775)	748 (449–1077)	0.040
HF/HR	125 (94–153)	114 (88–149)	0.683	LF/HF	4 (2–5)	3 (2–5)	0.783

Values are represented as median [interquartile range]. SDNN: Standard deviation of NN intervals; VLF: Very low frequencies; LF: Low frequencies; HF: Low frequencies; SU: Supine position; ST: Standing position; WP: Withdrawal phase; APP: Active pill-taking phase.

## Data Availability

The raw data supporting the conclusions of this article will be made available by the authors upon reasonable request. The data are not publicly available, as these data concern Olympic athletes.

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
