# Peer review of "Heart Rate Variability Measurements Across the Menstrual Cycle and Oral Contraceptive Phases in Two Olympian Female Swimmers: A Case Report"

_sports, 2025, doi:10.3390/sports13060185_

Round 1

Reviewer 1 Report

Comments and Suggestions for Authors

In general, the study was very well designed and is of great interest for practitioners.

I have some comments regarding the data collection, ,analysis and presentation.

The study the design must be better detailed. The figure provides a great idea on the study, but the written part is too short.

It is not clear what was assessed and where it is presented the data collected from the urinary ovulation tests and the salivary samples.

The training load variations over the different periods is a very important data, and considering the competitive level of the athletes, this must be better explored in the results section. Authors could demonstrate sRPE data in tables or figures as well.

Considering the high variability in the time-domain HRV data, RMSSD and SDNN must be presented in logarithmic transformed units (ln).

Author Response

In general, the study was very well designed and is of great interest for practitioners.

I have some comments regarding the data collection, analysis and presentation.

The study the design must be better detailed. The figure provides a great idea on the study, but the written part is too short.

Response: Indeed, we have chosen to detail each part of the monitoring in the corresponding method subsection, rather than after the announcement of Figure 1. However, we have added some sentences to detail.

Lines 105-110: “For Athlete 1, the menstrual status, training load and HRV data recorded via orthostatic test were monitored from the end of February 2023 to the end of the Olympic Games in August 2024. Her HRV night measurements were recorded since November 2023 when she received the Ōura® ring. For Athlete 2, two 28-day oral contraceptive (OC) cycles were also retrospectively identified to align with the period during which she was wearing the Ōura® device and all HRV data (orthostatic test and night measurements) were recorded.”

Lines 142-143: Athlete 1 received the device in November 2023, and Athlete 2 in December 2022.

It is not clear what was assessed and where it is presented the data collected from the urinary ovulation tests and the salivary samples.

Response: Indeed, we have not presented the data collected from the salivary samples. Concerning urinary test data, the tests used in this study were based on basic colorimetric urine strips assessing luteinizing hormone surge (as mentioned in the sentence lines 159-160: “Ovulation generally occurs 12 to 24 hours after luteinizing hormone surge detected by the test”.), which only provide a qualitative result (presence or absence of a test line), rather than quantitative values. As such, there were no numerical or graphical data to present. However, we have added some details as well as a figure in supplementary file to present the data collected during Athlete 1’s third menstrual cycle (Supplementary Figure 1).

Lines 158-159: “From the third to sixth cycles, she used urinary ovulation strip tests each morning starting from day ten until a positive result (i.e., luteinizing hormone surge) to detect ovulation day (step two).”

Lines 167-170: “In addition, saliva samples, a valid method to identify normal from abnormal hormonal cycles [38,39], were performed on the third cycle as described by Lafitte et al., (2024) to measure 17β-estradiol (E2), progesterone (P4) and free testosterone concentrations (pg.mL-1)”

Lines 216-217: “Saliva samples, collected during the third MC (Supplementary Figure 1), shown expected E2 and P4 fluctuation and confirmed ovulatory cycle [38,39].”

The training load variations over the different periods is a very important data, and considering the competitive level of the athletes, this must be better explored in the results section. Authors could demonstrate sRPE data in tables or figures as well.

Response: We agree with the reviewer. As we are limited in figure/table numbers in the manuscript, we have presented the weekly training load as percentage of maximum week for each athlete in supplementary figure (Supplementary Figure 2). We have mentioned the supplementary figure 2 in results section:

Lines 226-228 and 292-293: “Her weekly training load as percentage of maximum week is presented in Supplementary Figure 2.”

Considering the high variability in the time-domain HRV data, RMSSD and SDNN must be presented in logarithmic transformed units (ln).

Response: We agree with the reviewer that logarithmic normalization (LN) can be useful for data normalization. We have applied this approach in a previous study (see: https://doi.org/10.3390/app11178106). However, in the present case, we believe that:

  • Normalizing HRV data would not affect the statistical outcomes, as we are not comparing athletes with each other;
  • Providing absolute values is more relevant for practitioners, as we have done previously (see: https://doi.org/10.1055/a-0877-6981).

Reviewer 2 Report

Comments and Suggestions for Authors

First and foremost, I would like to commend the authors for their valuable contribution to this field of research. The study represents a rigorous longitudinal investigation, which undoubtedly required substantial effort. The primary objective of the present study was to examine heart rate variability responses in elite female Olympic swimmers. Below you can find my specific comments for your manuscript.

Specific comments

Introduction

Lines 75–76: Please consider adding one to two sentences to clarify the novelty of your study. Specifically, explain the significance of assessing heart rate variability (HRV) in two female athletes with natural and unnatural menstrual cycles, and how this contributes to the existing body of knowledge.

L79: Include the main hypothesis of your study.

Methods

Figure 1: Please enhance the clarity and visibility of the figure. The lettering appears too thumb and it is difficult to read."

Lines 131–137: Please specify which hand the swimmers wore the biometric ring. This information would enhance the methodology's clarity and reproducibility.

Lines 169–181: You report two different methods for assessing training load, both of which are recognized as reliable for evaluating athletic training. However, a key concern pertains to the training zones. Did both swimmers complete the same training volume (in terms of kilometers) during the sessions? If not, this discrepancy represents a methodological limitation and should be addressed and discussed accordingly in the manuscript.

Results

General comment: Please try to increase the clarity of all the figures included.

Table 2: It is recommended to reformat the table into a horizontal (landscape) orientation to improve readability and presentation clarity.

Conclusion

As this is a case study, it is important to propose directions for future research related to the topic. For instance, should heart rate variability (HRV) variables be monitored and controlled exclusively in elite female athletes, or could similar approaches be beneficial for broader athletic populations?

Author Response

First and foremost, I would like to commend the authors for their valuable contribution to this field of research. The study represents a rigorous longitudinal investigation, which undoubtedly required substantial effort. The primary objective of the present study was to examine heart rate variability responses in elite female Olympic swimmers. Below you can find my specific comments for your manuscript.

Response: We really thank the reviewer for the amount and the rigor of the work and the quality of the expertise. Please, find a point-by-point answer.

Specific comments

Introduction

Lines 75–76: Please consider adding one to two sentences to clarify the novelty of your study. Specifically, explain the significance of assessing heart rate variability (HRV) in two female athletes with natural and unnatural menstrual cycles, and how this contributes to the existing body of knowledge.

Response:  We have added two sentences.

Lines 76-79: “The assessment of HRV across the menstrual cycle may provide valuable insights into ANS activity during the different phases of the cycle. Such measurements could help elite athletes and their staff to manage training loads and recovery strategies more effectively.”

L79: Include the main hypothesis of your study.

Response: We have added our hypothesis Lines 82-84: We hypothesized that a rebound in parasympathetic activity might be observed during the menstrual phase and during the withdrawal phase of combined OC use.”

Methods

Figure 1: Please enhance the clarity and visibility of the figure. The lettering appears too thumb and it is difficult to read."

Response: We have improved the clarity and visibility of the figure increasing the letter size.

Lines 131–137: Please specify which hand the swimmers wore the biometric ring. This information would enhance the methodology's clarity and reproducibility.

Response: We have specified this information.

Lines 143-145: “The athletes were asked to wear the ring at the index of their dominant hand every night and as much as possible during the day except for training session.”

Lines 169–181: You report two different methods for assessing training load, both of which are recognized as reliable for evaluating athletic training. However, a key concern pertains to the training zones. Did both swimmers complete the same training volume (in terms of kilometers) during the sessions? If not, this discrepancy represents a methodological limitation and should be addressed and discussed accordingly in the manuscript.

Response: The two swimmers did not train under the same regimen and they did not complete the same training volume due to differences in their distance specializations. Since the purpose of the study was not to compare the athletes, we did not consider this difference a major methodological limitation, but rather an asset allowing to adapt to different training regimen. However, we have added the training load kinetics description of both swimmers in a supplementary file (Supplementary Figure 2). We also added a sentence in the Discussion (Lines 402-404): “Unfortunately, it was not possible to use the same training load quantification method for both swimmers due to the different training approaches of their respective coaches.”

Results

General comment: Please try to increase the clarity of all the figures included.

Table 2: It is recommended to reformat the table into a horizontal (landscape) orientation to improve readability and presentation clarity.

Response: We have increased letter size to improve the clarity. Thanks for the recommendation. We have reformatted the table 2.

Conclusion

As this is a case study, it is important to propose directions for future research related to the topic. For instance, should heart rate variability (HRV) variables be monitored and controlled exclusively in elite female athletes, or could similar approaches be beneficial for broader athletic populations?

Response: Based on the reviewer’s comment, we have now improved our conclusion section. Therefore, we have made some corrections to clarify that HRV monitoring is useful for everyone, not only (elite) female athletes. But the specificities of females could be explored and considered in a better way with HRV measurements.

Lines 437-443: “This case study highlights the importance of considering individual physiological responses when prescribing training and emphasizes the need for personalized approaches to optimize both performance and recovery. HRV is a practical and accessible tool that can be monitored daily using recent devices. Collecting long-term HRV data may provide deeper physiological insights across various populations including elite athletes. In particular, HRV monitoring could be especially valuable for exploring female-specific physiological characteristics.”
